# Facile Synthesis of the Composites of Polyaniline and TiO_2_ Nanoparticles Using Self-Assembly Method and Their Application in Gas Sensing

**DOI:** 10.3390/nano9040493

**Published:** 2019-03-30

**Authors:** Lei Gao, Changqing Yin, Yuanyuan Luo, Guotao Duan

**Affiliations:** 1School of Optical and Electronic Information, Huazhong University of Science and Technology, Wuhan 430074, China; lgao@issp.ac.cn; 2Key Lab of Materials Physics, Anhui Key Lab of Nanomaterials and Nanotechnology, Institute of Solid State Physics, Chinese Academy of Sciences, Hefei 230031, China; ycqdomain@163.com

**Keywords:** titanium dioxide, polyaniline, self-assembly, gas sensors, p–n junction

## Abstract

The composites of polyaniline and TiO_2_ nanoparticles with different contents were prepared in the aqueous solution of phosphoric acid, in which the phosphoric acid was selected as the protonic acid to improve the conductivity of polyaniline. In the composites, the TiO_2_ nanoparticles with the size of about 20 nm were coated by a layer of polyaniline film with a thickness of about 5 nm. Then, the gas sensors were constructed by a liquid–gas interfacial self-assembly method. The gas-sensing properties of the composites-based gas sensors obviously improved after doping with TiO_2_ nanoparticles, and the sensor response of the composites increased several times to NH_3_ from 10 ppm to 50 ppm than that of pure polyaniline. Especially when the mass ratio of TiO_2_ to aniline monomer was 2, it exhibited the best gas response (about 11.2–50 ppm NH_3_), repeatability and good selectivity to NH_3_ at room temperature. The p–n junction structure consisting of the polyaniline and TiO_2_ nanoparticles played an important role in improving gas-sensing properties. This paper will provide a method to improve the gas-sensing properties of polyaniline and optimum doping proportion of TiO_2_ nanoparticles.

## 1. Introduction

In recent years, industrial safety, environmental protection and public safety have been paid more and more attention [1]. Gas sensors, as sensing components, have been widely used in these fields such as chemical plant explosive detection and toxic gas leakage monitoring. They can monitor the composition and concentration of various toxic or harmful gases qualitatively or quantitatively in real time [2,3,4,5,6]. There are many kinds of gas sensors, for example catalytic combustion gas sensors, electrochemical gas sensors and thermal conductivity gas sensors and so on. Among them, semiconductor gas sensors are usually made by inorganic semiconductor metallic oxide materials for instance ZnO, SnO_2_, In_2_O_3_, Co_3_O_4_, WO_3_ and TiO_2_ etc. [7]. Generally speaking, they have the advantages of small size, light weight, easy integration and low cost, and are widely used in industrial hazardous gas leakage detection, toxic or harmful gas detection, flammable and explosive gas early warning and other fields [8,9]. However, for their applications in flammable or explosive gas detection, semiconductor gas sensors still have some shortcomings such as long response time, low sensitivity and especially high working temperature. The higher working temperature will bring higher energy consumption, shorten the work time and become a source of the explosion in the flammable or explosive gas detection. Therefore, it is necessary to develop a gas sensor operating at the lower temperature (even room temperature) and having excellent gas sensing performances such as high sensitivity, stability and selectivity etc.

Polyaniline (PANI), a p-type conductive polymer, has been widely used in metal anti-corrosion, drug delivery, electromagnetic shielding, secondary batteries, supercapacitors and other fields for its own unique advantages such as good environmental stability, low cost, low density, excellent flexibility and so on [10]. Especially for room temperature sensitivity and tunability of conductivity under the action of electrochemical redox and protonic acid, PANI is an ideal room temperature sensitive material and has been paying more and more attention in the field of gas sensors in recent years [11,12,13]. However, pure PANI sensors have some problems such as low sensitivity, high detection limit, slow response and long recovery time [14]. On the other hand, titanium dioxide (TiO_2_), a typical n-type semiconductor metallic oxide with a wide band gap (3.2 eV), exhibits excellent photocatalytic activity, stability, non-toxicity, low cost and unique physical and chemical properties and has also application in the areas of photocatalysis, solar cells and gas sensors etc. [15,16]. However, the TiO_2_ based gas sensors usually need a high temperature for operation, so it is desirable to optimize its gas sensing performances. The composites of PANI and TiO_2_ can combine the merits of both materials and have excellent gas sensing performances such as low operating temperature, high sensitivity and excellent reproducibility. Because the presence of p–n junction owing to the composition of PANI and TiO_2_ nanoparticles (NPs) will give rise to lower activation energy for easier adsorption of target gases. The p–n junction can also control the electric current flow in TiO_2_ and more electrons would be available for conduction [17,18,19,20]. The resulting composites will be superior to the PANI or TiO_2_ nanoparticles due to the complementary and synergistic effect. For example, Liu et al. [21] prepared PANI-TiO_2_-Au nanocomposite thin film on gold interdigital electrodes, and the detection limit to NH_3_ can reach to 1 ppm at room temperature and the gas response was 2.23–50 ppm NH_3_. Li et al. [22] via polymerization of aniline with electro-spun Mn_3_O_4_/TiO_2_ nanofibers prepared PANI/TiO_2_ composite nanofibers. At room temperature the composites had the detection limit for NH_3_ concentrations down to 25 ppb, which was a factor 10^−3^ decreased compared to that with the pure PANI fibers. From these results it can be seen that the composites of PANI and TiO_2_ have better gas-sensing properties. However, how to construct more excellent gas sensitive films and improve the compatibility of both materials is also an important factor for gas sensors.

Herein, using ammonium persulfate as oxidant the composites of PANI and TiO_2_ nanoparticles (namely TiO_2_@PANI) with different contents are prepared in the aqueous solution of phosphoric acid. The phosphoric acid is used as a kind of protonic acid, which can change the conjugate state of the polymer chain and improve the conductivity of PANI. The gas sensing test results show that the gas-sensing properties of the composites are better than those of pure PANI or TiO_2_ NPs. When the mass ratio of titanium dioxide to aniline monomer is 2, the gas sensors have the best gas-sensing properties. The ammonia (NH_3_) is selected as the target gas, which is a colorless and pungent gas and generated by animals and plants rotting, livestock farms, industrial nitrogen fixation and domestic refuse dump. NH_3_ gas is harmful to people’s health. When a small amount of ammonia is inhaled by people (10–15 ppm), it will usually cause rhinitis, pharyngitis, sore throat and hoarseness. In severe cases (above 10,000 ppm), hemoptysis, pulmonary edema, white or bloody sputum, dyspnea even death can be happened. So the achieving of NH_3_ gas monitoring and prediction is of great realistic significance and broad application prospects.

## 2. Experimental

### 2.1. Materials

Aniline (C_6_H_7_N, Ani), Ammonium persulfate ((NH_4_)_2_S_2_O_8_, APS), Phosphoric acid (H_3_PO_4_, PA), Sodium laurylsulfonate (C_12_H_25_SO_3_Na, SDS), methanol and ethanol were analytical reagent and purchased from Sinopharm Chemical Reagent limited corporation (Shanghai, China). Nano titanium dioxide powder (TiO_2_) with an average particle size of 20 nm was purchased from Alfa Aesar Chemical Reagent limited corporation (Shanghai, China). All of the chemical reagents were used directly without any further purification. Deionized water was obtained from a pure water device (Milli-Q, Millipore, MA, Molsheim, France).

### 2.2. Synthesis of TiO_2_@PANI

Firstly, 0.18 g Ani and 3.3 g SDS was dissolved in 10 mL deionized water and stirred for 15 min in an ice bath. Then a certain amount of TiO_2_ powder was added in the above solution and kept for 15 min, in which the mass ratio of TiO_2_ and Ani was 0, 0.5, 1, 2 and 5 (marked as m (TiO_2_)/m (Ani) = 0, 0.5, 1, 2 and 5), respectively. Next, 10 mL PA solution (0.1 mol/L) was added to the reaction solution and stirred for 30 min. Then 0.4568 g APS was dissolved in 10 mL deionized water and added into the mixed solution drop by drop after cooling to 0 °C. Then they were stirred for 24 h without stop in the ice bath. After reaction, the resulting precipitate was collected and washed several times with deionized water, ethanol and methanol, respectively. Finally, the obtained TiO_2_@PANI products were dried in a vacuum oven at 45 °C for 24 h.

### 2.3. Construction of Sensors by a Liquid–Gas Interfacial Self-Assembly Method

Here the gas sensors were constructed by a liquid–gas interfacial self-assembly method, and the details were as follows: Firstly, 10 mg as-dried TiO_2_@PANI products was dispersed in the 5 mL mixture solution of deionized water and ethanol with the volume ratio of 50% and treated with sonication for 60 min. Then the above 200 μL mixed solution was slowly dropped on the surface of deionized water along one side of a 500 mL beaker which was full of deionized water. In this process, the emerald green TiO_2_@PANI products spread out quickly and evenly on the surface of the water. Meanwhile the TiO_2_@PANI products would aggregate with each other under the thrust effect of ethanol. Then they self-assembly formed a thin film under the action of Van Der Waals forces only on the surface of the water. Last, the emerald green thin film was picked up by a piece of flat eletrode from the underside of liquid surface and dried in air at room temperature, and the gas sensors were fabricated. The flat electrode was interdigitated gold electrodes array on Al_2_O_3_ substrate with the size of 1.5 × 1 × 0.3 mm^3^, and the groove shape was like the letter “S”. Under the Al_2_O_3_ substrate, it was SiO_2_ dielectric insulation layer and ruthenium oxide heating wires. The bottom layer was silicon. 4 Pt wires were used to connect with testing system. In the experiments, the heating wires were not worked and all the gas sensing measurements were operated at room temperature. Meanwhile 10 mg TiO_2_ NPs were dispersed in 5 mL mixture solution of deionized water and ethanol with the volume ratio of 50% and the same operations were repeated again, then the gas sensors based on TiO_2_ NPs were obtained.

### 2.4. The Gas Sensing Measurements

All the gas sensing performances were measured in a static system, and the volume of the gas chamber was 20 L as shown in the Figure 1. Firstly, the prepared gas sensors were connected with the testing system by Pt wires according to the Figure 1. A 10 V regulated power (V_c_) was obtained by a DC power supply (Agilent U8002A, San Jose, CA, USA), and the value of load resistance (R_L_) was 10 Mohm. The output voltage (V_out_) of load resistance was collected by a multimeter (Agilent mod, U3606A, San Jose, CA, USA). By a simple calculation, the resistance changes of sensing devices (R_g_ and R_0_) were obtained, where R_g_ and R_0_ is the resistances of the sensor devices after and before exposure to target gas. In the process of testing, a certain amount of target gas was obtained from a standard NH_3_ cylinder with the concentration of 20,000 ppm, which was injected into the gas chamber by a syringe and diluted to the target concentration. The target concentration was calculated based on the injected amount of target gas. If 1 mL standard NH_3_ was injected into the gas chamber, the concentration of NH_3_ was 1 ppm. Then the resistance of sensing devices (R_g_) under different concentration of target gas was obtained. All the measurements were operated at room temperature (25 °C), and the heating voltage (V_h_) was always 0 V. Meanwhile, the relative humidity (RH) was remained at 60%.

### 2.5. Characterization Methods

The field-emission scanning electron microscopy (Hitachi SU8020, Tokyo, Japan) and the high-resolution transmission electron microscopy (JOEL JEM-2010, Tokyo, Japan) was used to observe the morphologies of products. The energy spectrum of product was obtained on the SU8020 with the attachment of energy dispersive X-ray spectroscopy (EDX, Tokyo, Japan). The acceleration voltage was 10.0 kV, and the selection area was 4.5 × 6.6 μm^2^. The information of N, O, Si, P and Ti element was obtained. The sample was firstly deposited onto the surface of the clean small silicon wafer before observing by SU8020, so the silicon in the energy spectrum corresponded to the silicon wafer substrate. A NICOLET MX-1E Fourier transformed spectrometer with KBr tablet (Thermo Fisher Scientific, Waltham, MA, USA) was used to measure the FTIR spectra of the products. The X-ray diffraction (XRD) was performed on a Philips X’Pert X-ray diffractometer (Almelo, The Netherlands) which the Cu-K line was 0.15419 nm.

## 3. Results and Discussion

### 3.1. Morphology

The morphology of products is observed by field-emission scanning electron microscopy (SEM). From Figure 2a, it is obvious that the pure PANI film prepared by the method of liquid phase oxidation without TiO_2_ has a porous structure, and some aggregation is observed. In the process of the PANI synthesis, mechanical agitation can prevent the aggregation of PANI. From Figure 2b, it can be seen that the TiO_2_ is easily agglomerated. The agglomerates are about 1.5 μm, and full ultrasonic treatment is needed before use. After adding TiO_2_ in the reaction system, the surface of TiO_2_ is coated by a layer of PANI film as shown in Figure 2c. This indicates that a good encapsulation effect on the surface of TiO_2_ can be achieved. From the SEM photograph of cross section in Figure 2d, it can be seen that the thickness of the film is about 1.5–2 μm. In addition, the film is formed by the accumulation of granular complex. It is relatively loose and porous, which is favorable to improve the properties of the gas sensors.

The transmission electron microscope (TEM) photograph of TiO_2_@PANI products at m (TiO_2_)/m (Ani) = 2 is shown in Figure 3a. It can be clearly seen that the TiO_2_ crystallites are about 20 nm in size. The spacing of the lattice stripe of TiO_2_@PANI is about 0.352 nm, which corresponds to the TiO_2_ (101) crystal plane of anatase in Figure 3b [23]. On the surface of TiO_2_ NPs polyaniline film can be seen and the thickness of the coating is about 5 nm, but there is no lattice stripe or diffraction pattern for this poor crystalline substance. Figure 3c is the selected area electron diffraction pattern of the TiO_2_@PANI, and it mainly also corresponds to the TiO_2_ (101) plane of anatase phase [24]. It also indicates that the TiO_2_ nanoparticles are polycrystalline particles. From the energy spectrum of the TiO_2_@PANI (measured by the SU8020, Figure 3d), it is visible that the products contain the elements of the PANI. The silicon in the energy spectrum corresponds to the silicon wafer substrate. These mean that the TiO_2_ NPs coated by PANI have been successfully prepared by the above method.

### 3.2. Characteristics Peaks

In order to demonstrate the characteristic peaks of the products, the FTIR spectrum of pure TiO_2_ NPs, the pure PANI and the TiO_2_@PANI samples at m (TiO_2_)/m (Ani) = 2 is measured, and the results are shown in Figure 4. In the FTIR spectra of pure TiO_2_ NPs (Figure 4a), only three main absorption peaks are found. The wide peak at the 660 cm^−1^ is caused by the stretching vibration of the Ti-O bond in the NPs [25]. The sharp absorption peak near the 1639 cm^−1^ is due to the characteristic absorption peak of water in TiO_2_ NPs, which adsorbs the water of the air [26]. The broad peak at 3450 cm^−1^ is caused by the stretching vibration of hydroxyl groups [27]. In Figure 4b,c, several characteristic absorption peaks appeared; the peak at 1143 cm^−1^ corresponds to N=C stretching mode of the quinoid units [28]. The absorption peak near 1304 cm^−1^ is C-H stretching vibration of the benzene ring [29], and at the 823 cm^−1^ it is the C-H outer bending vibration on the symmetric substitution of the benzene ring [30]. At 1480 and 1384 cm^−1^ it is the C=C stretching vibration of the benzene ring, and the peak at 1127 cm^−1^ is associated with quinonoid unit. The peak at 1036 cm^−1^ is due to in-plane vibration of C-H bending mode [31]. The appearance of these peaks indicates that the products have the molecular structure of PANI. Comparing the infrared absorption spectra of TiO_2_@PANI and pure TiO_2_ NPs, it is found that the absorption peak near 660 cm^−1^ is weaker for the TiO_2_@PANI, while the other peaks are largely unchanged. That is because the surface of TiO_2_ NPs are coated by PANI. At the same time, the characteristic absorption intensity of PANI (1127, 1036 and 823 cm^−1^) is also weakened for the TiO_2_@PANI. The reason is that there is a strong interaction between PANI and TiO_2_ NPs. The strong interaction may be associated with the interaction of titanic and nitrogen atom in PANI. As titanium is a transition metal, it has intense tendency to form coordination compound with nitrogen atom in PANI. This interaction may weaken the bond strengths of PANI. Moreover, the action of hydrogen bonding between TiO_2_ NPs and PANI is also contributory to this weakness of bands [32].

Figure 5 is the XRD diffraction pattern of the above three samples. In Figure 5a, the major peaks in the XRD pattern are in agreement with standard JCPDS card of tetragonal anatase (No. 21-1272) [33]. The strong and sharp peak types and the smooth baseline indicate a good crystallinity. However, in Figure 5b for the XRD diffraction pattern of PANI film, only a broad reflection peak appears at about 23°, indicating an amorphous pattern. Figure 5c is the XRD diffraction pattern of the TiO_2_@PANI sample at m (TiO_2_)/m (Ani) = 2. It is obvious that almost the same sharp diffraction peaks appear at the corresponding position of pure TiO_2_ NPs. However, the intensity of the diffraction peak becomes weaker, which may be the result of the existence of PANI coating [34]. In addition, according to the Scherrer formula:D = Kλ/βcosθ
where D is crystallite size, K is a constant (K = 0.89), λ is X ray wavelength, and β is the full width at half maximum. From the diffraction peaks of TiO_2_ NPs we can calculate that the crystallite size of TiO_2_ NPs is about 20 ± 3 nm, which is consistent with the results of HRTEM.

### 3.3. Gas-Sensing Properties

The gas sensors are fabricated with different m (TiO_2_)/m (Ani) ratios from 0 to 5, and the gas sensing performances of these sensors to NH_3_ gas are measured at room temperature. The sensor response curves to NH_3_ gas ranging from 10 to 50 ppm are shown in Figure 6a. Here, the sensor response is defined as R_g_/R_0_, where R_g_ and R_0_ is the resistances of the gas sensors after and before exposure to NH_3_ gas. It can be seen that the sensor response of TiO_2_@PANI samples is obviously improved comparing with that of PANI. As the content of TiO_2_ nanoparticles increases to a certain amount, the sensor responses of TiO_2_@PANI samples also increase. When the ratio of m (TiO_2_)/m (Ani) reaches 2, the sensor response reach a maximum. It is about 11.6 to 50 ppm NH_3_ gas. However, with the continued increasing of the ratio of TiO_2_ nanoparticles to 5 the sensor response obviously decreases. There also is a very good linear relationship between the sensor response and the concentration of NH_3_ (Figure 6b), which is favor to the practical application of the sensors. When the gas sensor devices are exposed to different contents of NH_3_ gas, the resistances of the gas sensor are rapidly increasing and will be stable after a certain time. When they are exposed to air, the recovery of the gas sensors is also fast and gradually approaching the baseline. The response/recovery times of the gas sensors to different contents of NH_3_ gas can be seen in the Table 1. The response time (defined as the time for reaching 90% of the full response of the sensor after the gas sensor is exposed to target gas) of gas sensor with m (TiO_2_)/m (Ani) = 2 to 50 ppm NH_3_ gas is about 26 s, and the recovery time (defined as the time for falling to 10% of the maximum response after the target gas gets out) is 142 s. It is found that the addition of TiO_2_ nanoparticles almost has no effect on the response or recovery time of the gas sensors, even so the sensor response is obviously improved. In Table 2, the gas-sensing properties of the PANI/TiO_2_ composites are compared with the reported gas sensors in recent publications [22,35,36,37,38,39]. The results show that the gas sensor prepared by a liquid–gas interfacial self-assembly method at m (TiO_2_)/m (Ani) = 2 exhibits excellent sensor response and low response time to NH_3_ gas at room temperature. In addition, the sensor response of TiO_2_ NPs to NH_3_ gas ranging from 10 to 50 ppm is measured at room temperature, and the result is shown in Figure 7. It can be seen that at room temperature the TiO_2_ NPs have no response to NH_3_ gas, and the value of the sensor response is about 1.0 except for a little fluctuation. The sensor response of TiO_2_ NPs to other gases or at high temperature will be investigated in the further.

The repeatability is also a very important index for gas sensors. So the repeatability response curves of TiO_2_@PANI samples with m (TiO_2_)/m (Ani) = 2 are measured under 50 ppm ammonia as shown in Figure 8. It can be seen that the sensor response after five repeats is 10.7, 11.0, 10.8, 11.2 and 12.0 at room temperature. There are little noticeable changes, which means that the gas sensors have a good stability for NH_3_ gas and it is favorable for industrial applications.

Including the sensor response and repeatability, the selectivity of gas sensors is also an important index. Here, six different kinds of gases such as nitrogen dioxide (NO_2_), ethanol (C_2_H_6_O), acetone (C_3_H_6_O), hydrogen sulfide (H_2_S), methane (CH_4_) and sulfur dioxide (SO_2_) are selected as the interference gases. In Figure 9a, it can be seen that the sensor response to 50 ppm ammonia, 50 ppm nitrogen dioxide, 50 ppm ethanol, 50 ppm acetone, 50 ppm hydrogen sulfide, 50 ppm methane and 50 ppm sulfur dioxide is 11.3, 3.0, 1.1, 1.4, 0.9, 1.0 and 1.2 at room temperature, respectively. There is a significant difference between the interfering gases and the target gas. The sensor shows weak response to the interfering gases. The reason is that only NH_3_ gas has the interaction with the protons in the PANI, and the interfering gases have little interaction with the PANI or TiO_2_ NPs. At high temperature these interfering gases may have interaction with the PANI or TiO_2_ NPs and it will be investigated in the further. In view of an industrial application, the influence of relative humidity (RH) should be considered. All the above experimental results are obtained under 60% of RH at room temperature. Now the value of RH is changed from 20 to 80% as shown in Figure 9b. It can be seen that under different relative humidity conditions the sensor response with m (TiO_2_)/m (Ani) = 2 is 21.8, 17.7, 11.8 and 8.2 to 50 ppm NH_3_ gas. At 80% RH, it has the worst performance. The reason is that as the value of RH increases, the active sites of gas sensor are occupied by more water, which results in the decrease of detection sites to NH_3_ gas, and the performances of the gas sensor becomes worse.

From the above results, it can be seen that when the ratio of m (TiO_2_)/m (Ani) is 2, the gas sensor device has the best gas sensing responses, repeatability and selectivity. Firstly, the reaction between PANI and NH_3_ is a reversible protonation–deprotonation process [40]. When NH_3_ is adsorbed into the polymer chains, the conjugation of the polymers is destroyed and results in the increase of the resistance. When NH_3_ is taken away, the resistance of PANI will gradually recover. As the concentration of NH_3_ increases, the amount of adsorbed NH_3_ increases and the resistance of gas sensor increases obviously. On the other hand, we can see that the composites of PANI and TiO_2_ NPs have more excellent sensor responses comparing with the pure PANI, and this phenomenon may be explained by the synergies effect and complementary behaviors of PANI and TiO_2_ NPs [41]. Figure 10 exhibits a schematic energy diagram to further illuminate the gas sensing mechanism of PANI/TiO_2_ composites. The energy between the highest occupied molecular orbital (HOMO) and the lowest unoccupied molecular orbital (LUMO) of PANI is 2.8 eV, and the energy between valence band (VB) and conduction band (CB) of TiO_2_ is 3.2 eV. Because the energy band gap matches well between the LUMO level of PANI and the CB of TiO_2_, the charge separation is enhanced. Such enhancement promotes the gas sensing ability of the PANI/TiO_2_ composites to NH_3_ [42]. Meanwhile the sensing performance of the composites with different amounts of TiO_2_ NPs is different. It relates to the p–n junction of PANI/TiO_2_ composites [22]. The absorption of ammonia will not only change the conductivity of PANI but also the resistance of p–n contact interface. The different proportion of TiO_2_ NPsof the composites determines the channels of electronic transmission. For the pure PANI, the conductive channel is along the polymer chains, but for the composite, the conductive channel is different. When the proportion of TiO_2_ NPs is low, the PANI layer is the main conductive channel. With the increase of TiO_2_ NPs in a certain range, the area of the p–n junction interface increases and gradually becomes the main conductive channel and the gas sensor has a better response. When the proportion of TiO_2_ NPs is too high, partly TiO_2_ NPs will become the conductive channel resulting in the decrease of the sensor response [22,43]. When the mass ratio of m (TiO_2_)/m (Ani) is 2, it has the best proportion under the experimental condition.

## 4. Conclusions

In summary, the composites of PANI and TiO_2_ nanoparticles were successfully prepared in the aqueous solution of phosphoric acid, in which phosphoric acid was used as dopant. The granular TiO_2_ nanoparticles were about 20 nm in size and coated by a layer of PANI film on their outer surface. Gas sensors with different contents of TiO_2_ nanoparticles were constructed by a liquid–gas interfacial self-assembly method and exhibited better gas-sensing properties than pure PANI to NH_3_ from 10 ppm to 50 ppm. The composites with m (TiO_2_)/m (Ani) = 2 exhibited the best gas sensitivity (about 11.2 to 50 ppm), stability and selectivity to NH_3_ at room temperature. The formation of a p–n junction in the composites was one of the important factors for excellent gas-sensing properties. It provided a preparation method of a gas sensor for detecting nitrogen concentration at room temperature with good gas-sensing properties.

## Figures and Tables

**Figure 1 nanomaterials-09-00493-f001:**
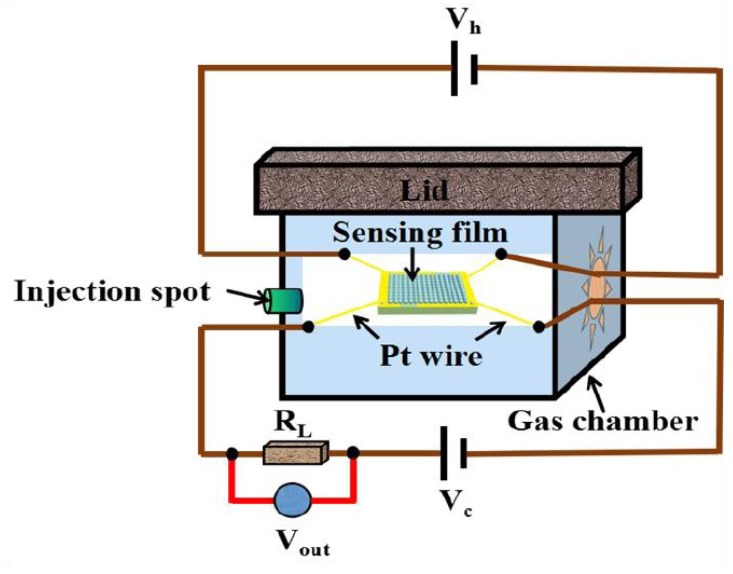
The gas sensitive testing system.

**Figure 2 nanomaterials-09-00493-f002:**
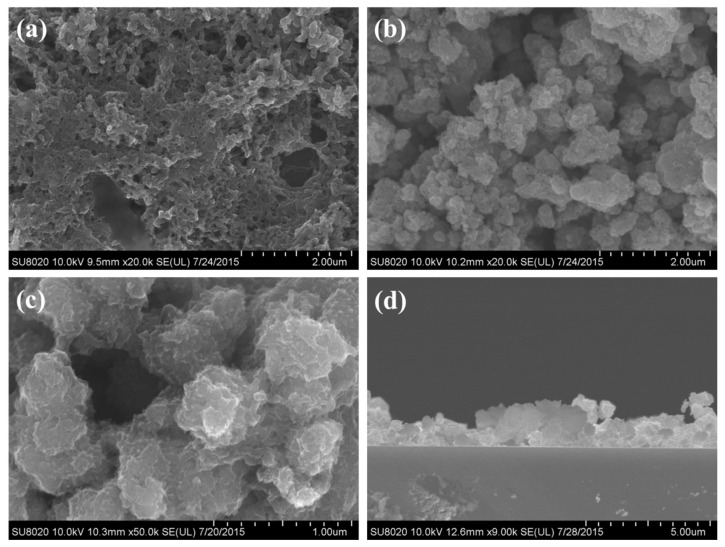
SEM images of (**a**) the pure PANI; (**b**) pure TiO_2_ NPs; (**c**) TiO_2_@PANI products at m (TiO_2_)/m (Ani) = 2 and (**d**) cross section of gas sensing thin film prepared by the liquid–gas interfacial self-assembly method at m (TiO_2_)/m (Ani) = 2.

**Figure 3 nanomaterials-09-00493-f003:**
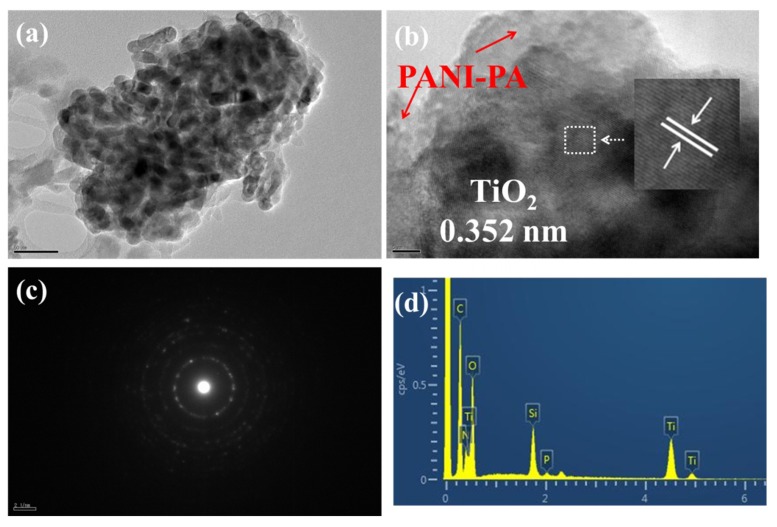
(**a**) TEM image of the TiO_2_@PANI product at m (TiO_2_)/m (Ani) = 2 and the scale is 50 nm in the lower-left corner; (**b**) HRTEM image of the product and the scale is 5 nm, and the insert picture is the enlarged picture of the lattice stripe of TiO_2_ NPs; (**c**) the selected area electron diffraction and (**d**) the energy spectrum of the TiO_2_@PANI sample.

**Figure 4 nanomaterials-09-00493-f004:**
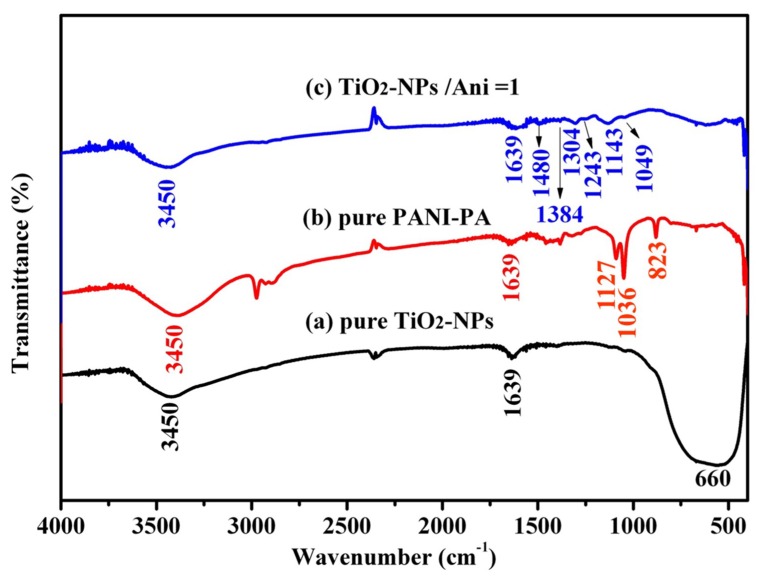
FTIR spectrum of (**a**) pure TiO_2_ NPs; (**b**) the pure PANI and (**c**) TiO_2_@PANI products at m (TiO_2_)/m (Ani) = 2.

**Figure 5 nanomaterials-09-00493-f005:**
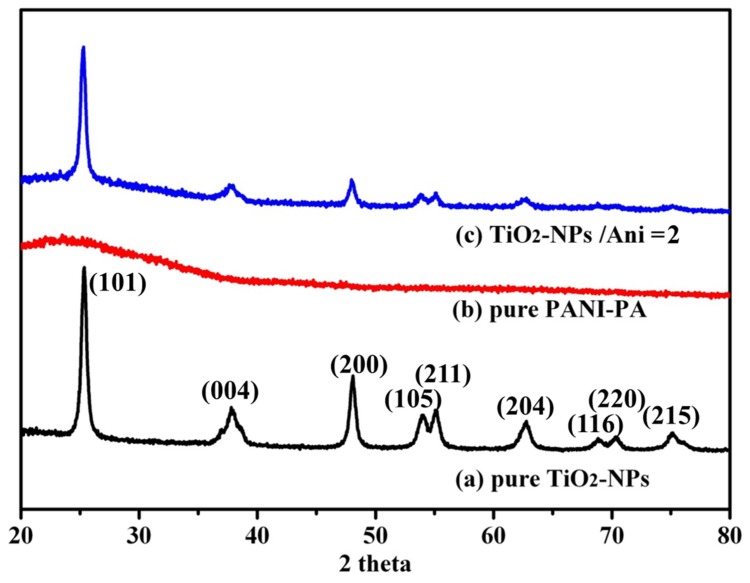
XRD pattern of (**a**) pure TiO_2_ NPs; (**b**) the PANI films prepared by phosphoric acid and (**c**) TiO_2_@PANI sample produced at m (TiO_2_)/m (Ani) = 2.

**Figure 6 nanomaterials-09-00493-f006:**
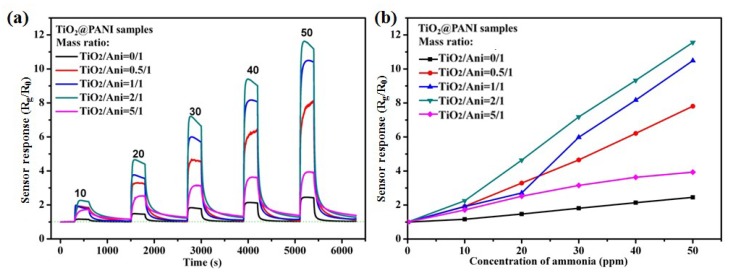
(**a**) Sensor response of the TiO_2_@PANI samples with different m (TiO_2_)/m (Ani) to NH_3_ gas ranging from 10 to 50 ppm at room temperature; (**b**) the corresponding sensitivity curve.

**Figure 7 nanomaterials-09-00493-f007:**
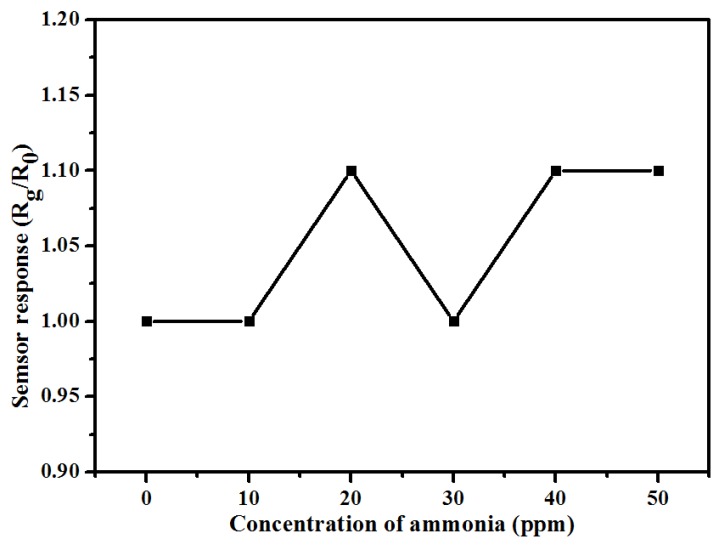
Sensor responses of TiO_2_ NPs to NH_3_ gas ranging from 10 to 50 ppm at room temperature.

**Figure 8 nanomaterials-09-00493-f008:**
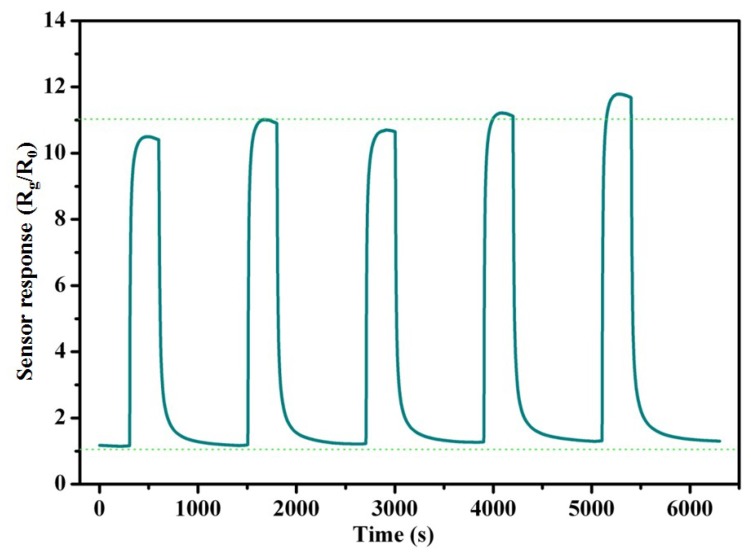
The repeatability curve of TiO_2_@PANI samples with m (TiO_2_)/m (Ani) = 2 to 50 ppm NH_3_ gas at room temperature, and the green dash line (below) is the baseline and the upper green dash line represents the average value of the sensor responses.

**Figure 9 nanomaterials-09-00493-f009:**
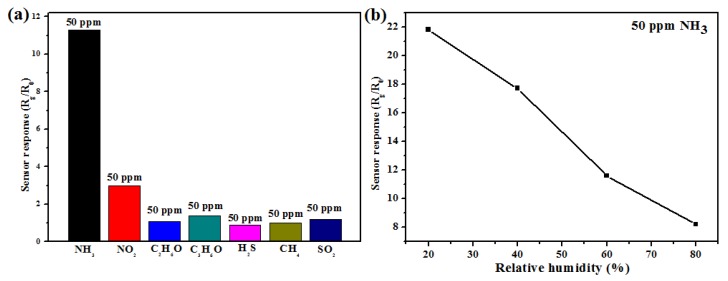
(**a**) The sensor responses of TiO_2_@PANI samples with m (TiO_2_)/m (Ani) = 2 to NH_3_ (50 ppm), NO_2_ (50 ppm), C_2_H_6_O (50 ppm), C_3_H_6_O (50 ppm), H_2_S (50 ppm), CH_4_ (50 ppm) and SO_2_ (50 ppm) at room temperature; (**b**) The influence of relative humidity to the performances of sensor to 50 ppm NH_3_ at room temperature (between 20 and 80% of RH).

**Figure 10 nanomaterials-09-00493-f010:**
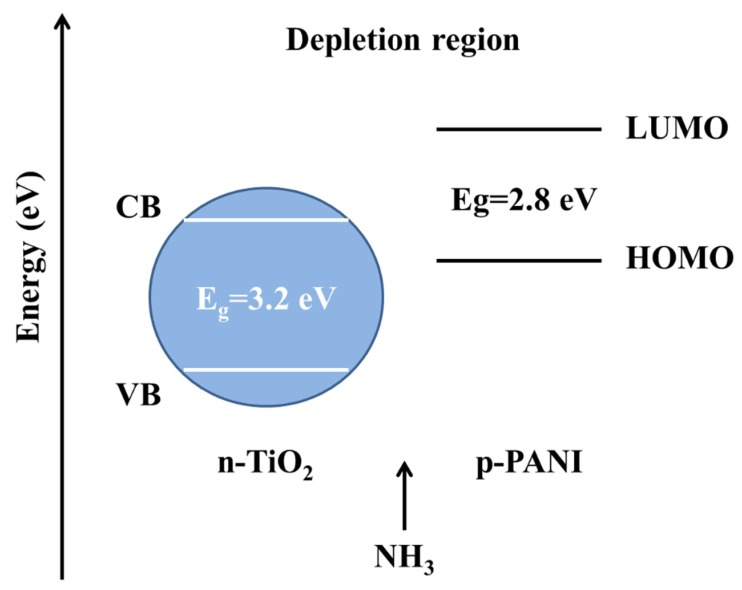
The schematic energy diagram of the composites of PANI and TiO_2_ NPs.

**Table 1 nanomaterials-09-00493-t001:** The response/recovery times of the gas sensors to different contents of NH_3_ gas, where the r is the mass ratio of TiO_2_ and Ani (m (TiO_2_)/m (Ani)).

	r	0	0.5	1	2	5
NH_3_						
10 ppm		12/55	13/222	45/378	62/451	73/728
20 ppm		17/50	18/173	26/246	31/342	66/620
30 ppm		19/50	22/104	29/199	28/246	65/522
40 ppm		21/49	32/90	43/188	27/180	63/435
50 ppm		22/48	35/85	55/170	26/142	61/333

**Table 2 nanomaterials-09-00493-t002:** Summary of recent publications about the NH_3_ detection based the PANI/TiO_2_ composites.

Materials	Detection Limit	Response/NH_3_ Concentration	Response Time (s)	Recovery Time (s)	References
PANI/TiO_2_/Au	1 ppm	2.23/50 ppm	122	-	[21]
PANI/TiO_2_	0.5 ppm	5.4/100 ppm	100	232	[35]
PANI/TiO_2_/PA6	-	2.6/250 ppm	150	450	[36]
PANI/TiO_2_/PA6	-	18.3/250 ppm	250	-	[37]
PANI/TiO_2_	-	48%/100 ppm	40	70	[38]
PANI/TiO_2_	20 ppm	47%/100 ppm	>40	>70	[39]
PANI/TiO_2_	25 ppm	0.8/200 ppb	80	-	[40]
PANI/TiO_2_	10 ppm	11.6/50 ppm	26	142	This Work

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
