# Peer review of "Facile Synthesis of the Composites of Polyaniline and TiO2 Nanoparticles Using Self-Assembly Method and Their Application in Gas Sensing"

_nanomaterials, 2019, doi:10.3390/nano9040493_

Round 1
Reviewer 1 Report
Dear Authors,
the English language must be carefully checked as too many sentences are not clear at all.
In addition, the following corrections have to be done:
* the introduction must be improved as PANI-TiO2 sensors for ammonia detection have been already described in the literature.
* Line 107: please, indicate the nature of the substrate and of the electrodes.
* Line 131: the sentence is not correct: the powder is agglomerated and there are agglomerates of about 1.5 micron, considering your 2 microns scale bar...
* Line 145: "black granular TiO2 NPs" are "TiO2 crystallites", please change the text.
* Line 153: Why is silica present in the EDX spectrum?
* Line 192-193: Scherrer's equation allows to determine crystallite size, not grain size.
+ you should use more than one peak and determine the standard deviation, not only the average value
* § 3.3 Gas sensing properties: You should compare your results with those of the existing literature on this topic to show the improvements of your work.
* Line 212: You should determine the sensitivity of your sensor, that is to say the slope of the calibration curve.
* Line 218: please, determine response and recovery times for the investigated sensors' compositions and gas concentrations and add a table with these values.
* Figures 5 & 6: On Y-axis: Sensor response, not gas response.
* Line 237 & 238: you determined the sensor response, not its sensitivity.
* Figure 7: On Y-axis: Sensor response, not sensitivity.
* The gas sensing mechanism part should be improved: illustrate the role of ammonia, please.
* As the aim is to develop a new sensing material in view of an industrial application, the interferences with humidity should be checked: please show the sensor response under different relative humidity levels.
Best regards.
Reviewer

Author Response
Answer: Thanks for your efforts in reviewing our manuscript. We are very grateful that you have raised some valuable comments leading to the improvements of the paper. According to your suggestions, the article has been modified carefully and the revised places are marked with red letters in the revised manuscript. The m(TiO2)/m(Ani)=2 was also marked in § 2.2. We hope that you will be satisfied with our revisions.

Reviewer 2 Report
See separate PDF-document

Author Response
Thanks for your efforts in reviewing our manuscript. We are very grateful that you have raised some valuable comments leading to the improvements of the paper. According to your suggestions, the article has been modified carefully and the revised places are marked with red letters in the revised manuscript. The numerous unusual and ungrammatical wordings have been corrected. We hope that you will be satisfied with our revisions.

Round 2
Reviewer 1 Report
Dear Authors,
thank you for having accepted the suggested corrections. However, there are still some sentences that still need a revision because of the English language. Thus, I encourage you to revise carefully all the text.
Please, find below some corrections, but the list is not exhaustive at all:
* Lines 26,27: "Meanwhile... properties". The sentence is a repitition of lines 22, 23 and can be deleted.
* Line 65: "titanium dioxide" and not "Titanium Dioxide".
* Line 188: "...it can be seen that the thickness of the film is...".
* Line 226 "...have appeared" or "appeared"".
* Line 230: "...of the benzene ring".
* Line 239: Caption of figure 4: "(b) refers to pure PANI-PA, not to the composite sensor".
* Line 256: β is the Full width at half maximum (FWHM).
* Lines 265, 267, 269, 291, 301: "sensor response" in place of "gas sensitive response" + Line 265 delete "of the sensors".
* Line 272: "reaches 2", not "reaches to 2".
* Line 278, 279: check the sentence, please.
* Line 292: replace "based" with "of".
* Lines 301-303: check the sentence, please.
* Lines 315,316: Check the caption of Fig. 7.
* Line 322: "means".
* Line 388: "becomes".
* Line 407: "...was one of the important factors for the excellent...".
Best regards.
Reviewer
Author Response
Thanks again for your efforts in reviewing our manuscript. We are very grateful that you have found some corrections. We have revised carefully all the manuscript, and all the revised places are marked with red letters in the new revised manuscript. The unusual wordings have been modified. Later the proof reading services will be provided from MDPI, and the charge will be paid.

Reviewer 2 Report
The quality of the manuscript has imroved significantly after revision by the authors. Although most of the suggestions and comments were addressed in the revised version of the manuscript by the authors, few comments have not been answered in a satisfying way:
1) I again would strongly recommend to use e.g. the proof reading offer (e.g.from MDPA) to reduce the unusual wordings which still appear in the text.
2) Although the suggsted reference of Li et al. has been added, a convincing argument, why the combination of TiO2 particles with PANI is so interesting is not given so far. The synergetic effect by the combination of TiO2 and PANI is not a new aspect. In this context the authors should correct the information that Li at al used PANI/Mn3O4/TiO2 composite fibers, as the Mn3O4 was used to form PANI from Ani, so the Mn component is consumed by the polymerization and a pure TiO2 fibers/PANI composit results. As the particle-based material has the detection limit which is a factor 10^-3 decreased compared to that with the fibers, the cheaper and less time consuming preparation of the particular system is not a convincing argument. Maybe it would be possible to give some outlook in how far the sensitivity of the particulate system might be increased to become compatititve with the fiber-based material in future.
3) In Figure 4 the labels for the double bond between 1200 cm^-1 and 1000 cm^-1 were deleted, however it is now not clear what is the origin of these two very strong bands, which appear in the red line (pure PANI) but not in in the sprectrum of the composite (blue line). This should be explained.
4) I appreciate the measurements, which wre carried out with pure TiO2 particles without PANI. As this is a new experiment, the preparation of the sensor which consists of TiO2 particles without PANI should be implemented in the experimental part.
Author Response
Thanks again for your efforts in reviewing our manuscript. We are very grateful that you have once more put forward some valuable comments leading to the improvements of the paper. According to your suggestions, the article has been modified carefully and the revised places are marked with red letters in the new revised manuscript. We hope that you will be satisfied with our revisions.
